# Surrogate-Assisted Automatic Parameter Adaptation Design for Differential Evolution

**Vladimir Stanovov *** and **Eugene Semenkin**

Institute of Informatics and Telecommunication, Reshetnev Siberian State University of Science and Technology, 660037 Krasnoyarsk, Russia; eugenesemenkin@yandex.ru
* Correspondence: vladimirstanovov@yandex.ru

**Abstract:** In this study, parameter adaptation methods for differential evolution are automatically designed using a surrogate approach. In particular, Taylor series are applied to model the searched dependence between the algorithm's parameters and values, describing the current algorithm state. To find the best-performing adaptation technique, efficient global optimization, a surrogate-assisted optimization technique, is applied. Three parameters are considered: scaling factor, crossover rate and population decrease rate. The learning phase is performed on a set of benchmark problems from the CEC 2017 competition, and the resulting parameter adaptation heuristics are additionally tested on CEC 2022 and SOCO benchmark suites. The results show that the proposed approach is capable of finding efficient adaptation techniques given relatively small computational resources.

**Keywords:** numerical optimization; differential evolution; parameter adaptation; surrogate assisted

**MSC:** 68W50; 68T20; 65K10; 90C59

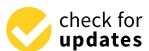



## 1. Introduction

In the area of evolutionary computation (EC), parameter adaptation and control is among the most-discussed topics due to the large number of values to be tuned and the high sensitivity of these values. For example, even a relatively simple genetic algorithm (GA) requires setting of population size, mutation rate and parameters controlling selective pressure. For most cases, there exist quite reliable recommendations about setting specific parameter values; however, better efficiency is mostly achieved through parameter adaptation techniques. One of the goals of parameter adaptation is to replace existing parameters with new ones, which determine adaptation but have a lower sensitivity and less effect on the final result.

Among EC methods for numerical optimization, differential evolution (DE) is one of the most popular nowadays [1,2]. The reason for the popularity of DE is that the classical algorithm has only three main parameters: population size, crossover rate and scaling factor. However, they should be carefully tuned to achieve better results. Many adaptation techniques have been proposed, including jDE-based [3] approaches and JADE [4] and SHADE-based [5] approaches, as well as many others. However, despite their high efficiency and various real-world applications, there still seems to be room for further efficiency improvement and simplification. One of the ways to develop new adaptation methods could be the usage of a hyper-heuristic (HH) approach, i.e., automatic development of heuristics.

In this study, heuristics are designed for scaling factor, crossover rate adaptation and population size control. For scaling factor and crossover rate, the current success rate value, i.e., the ratio of the number of improved solutions in every generation, is applied. This choice is inspired by a recent study [6] in which genetic programming was used as a hyper-heuristic approach. In the current study, instead of genetic programming, the usage of Taylor series is proposed. In particular, the series are used to find the shape of the

curve, describing the dependence between success rate and scaling factor or success rate and crossover rate. For population size, the dependence is searched between current computational resources and the population size reduction rate. The recently proposed L-NTADE [7] algorithm is used as a baseline approach in this study. To find the best coefficient in a Taylor series, the Efficient Global Optimization (EGO) algorithm is applied, which is based on a surrogate-assisted approach and uses Gaussian process to build a model of the target function. The experiments performed in the Congress on Evolutionary Computation (CEC) competition on numerical optimization 2017 [8], CEC 2022 [9] and SOCO [10], show that applying EGO allows novel parameter adaptation techniques for DE to be found, which may perform better than existing methods. The main features of this study can be outlined as follows:

1. The success rate value is a valuable source of information for determining the location parameter for sampling scaling factor values;
2. The Efficient Global Optimization algorithm is capable of determining the coefficients in a Taylor series with a relatively small number of evaluations;
3. The dependence between crossover rate and success rate is minor;
4. The designed population size control strategy is similar to linear population size reduction (LPSR), which proves the efficiency of classical LPSR;
5. The designed heuristics for scaling factor adaptation are capable of outperforming success history adaptation when used in a different algorithm or dimension or tested on a different benchmark suite, thereby showing generalization capabilities.

The rest of this paper is organized as follows. In Section 2, background studies are described. In Section 3 related works are considered. In Section 4, the proposed approach is presented. Section 5 contains the experimental setup and results, followed by the discussion in Section 6 and conclusions in Section 7.

## 2. Background

### 2.1. Differential Evolution

Differential evolution is a population-based numerical optimization method proposed in [11]. DE starts by initializing a set of $N$ individuals $x_i = (x_{i,1}, x_{i,2}, \ldots, x_{i,D})$, $i = 1, \ldots, N$, where $D$ is the dimension of the search space. In most cases, random initialization with uniform distribution is applied:

$$x_{i,j} = x_{lb,j} + rand \times (x_{ub,j} - x_{lb,j}). \tag{1}$$

where $j = 1, \ldots, D$. After evaluating all the solutions using the target function ($f(x)$), the main loop containing mutation, crossover and selection is started.

The difference-based mutation is the main feature of DE. There are several known mutation strategies; however, most modern approaches use *current-to-pbest/1*:

$$v_{i,j} = x_{i,j} + F \times (x_{pbest,j} - x_{i,j} + x_{r1,j} - x_{r2,j}), \tag{2}$$

where $v_i$ is the donor vector, $x_i$ is the target vector, $F$ is the scaling factor parameter, *pbest* is the index one of the top $p\%$ of individuals, and $r1$ and $r2$ are randomly chosen indexes so that $i \neq pbest \neq r1 \neq r2$. DE is known to be highly sensitive to the $F$ parameter [12], and most adaptation techniques are developed to tune it.

The donor vector ($v_i$) is further used in mutation, where it is combined with the target vector ($x_i$) to generate a trial vector ($u_i$) as follows:

$$u_{i,j} = \begin{cases} v_{i,j}, & \text{if } rand(0,1) < Cr \text{ or } j = jrand \\ x_{i,j}, & \text{otherwise} \end{cases}. \tag{3}$$

where $Cr$ is the crossover rate value ($\in [0,1]$), and *jrand* is a randomly chosen index from $[1, D]$ that is required to make sure that the trial vector is different from the target vector.

After crossover, the generated solution should be checked to ensure that it falls within the search space boundaries ($[x_{lb,j}, x_{ub,j}]$, $j = 1, \ldots, D$). One of the widely used bound constraint-handling methods is the midpoint target method, which works as follows:

$$u_{i,j} = \begin{cases} \frac{x_{lb,j} + x_{i,j}}{2}, & \text{if } v_{i,j} < x_{lb,j} \\ \frac{x_{ub,j} + x_{i,j}}{2}, & \text{if } v_{i,j} > x_{ub,j} \end{cases}. \tag{4}$$

If a coordinate of the trial vector violates a boundary, then the target vector coordinate is used to move towards the boundary.

The last step is called selection, although it is different from selection in genetic algorithms and plays a role of the replacement mechanism. If the trial vector's target function value ($f(u_i)$) is better than that of $x_i$, then the replacement occurs:

$$x_i = \begin{cases} u_i, & \text{if } f(u_i) \leq f(x_i) \\ x_i, & \text{if } f(u_i) > f(x_i) \end{cases}. \tag{5}$$

Such selection scheme is simple and efficient, as it allows for exploration of different areas of the search space. However, there have been known attempts to improve it [13].

### 2.2. Parameter Adaptation in Differential Evolution

One of the main advantages of DE, i.e., a small number of parameters, comes with a disadvantage: high sensitivity, especially to the scaling factor ($F$). Designing an efficient parameter adaptation scheme for DE could be challenging [14,15], and in most cases, one of the well-known methods is applied. The most widely used method is success–history adaptation (SHA), as proposed in the SHADE algorithm [5], which is based on JADE [16].

In SHA, the $F$ and $Cr$ values are sampled before every mutation and crossover using Cauchy and normal distribution. The location parameters for sampling are set to $M_{F,h}$ and $M_{Cr,h}$, and the scale parameter is set to 0.1. Here $M_{F,h}$ and $M_{Cr,h}$ are the values from the memory cells, containing a pair for $F$ and $Cr$, respectively. The number of sells is $H$, and $h$ is a random integer $\in [1, D]$.

The values of $F$ and $Cr$ which resulting in successful offspring are stored in $S_F$ and $S_{Cr}$. The improvement value ($\Delta f = |f(u_j) - f(x_j)|$) is also stored in $S_{\Delta f}$. At the end of every generation, one of the memory cells with index $k$ is updated as follows:

$$\begin{cases} M_{F,k}^{t+1} = 0.5(M_{F,k}^t + mean_{wL,F}) \\ M_{Cr,k}^{t+1} = 0.5(M_{Cr,k}^t + mean_{wL,Cr}) \end{cases}, \tag{6}$$

where $mean_{wL}$ is a weighted Lehmer mean, calculated as:

$$mean_{wL} = \frac{\sum_{j=1}^{|S|} w_j S_j^2}{\sum_{j=1}^{|S|} w_j S_j}, \tag{7}$$

where $w_j = \frac{S_{\Delta f_j}}{\sum_{k=1}^{|S|} S_{\Delta f_k}}$, $S$ is either $S_{Cr}$ or $S_F$. The index ($k$) of a memory cell to be updated is iterated every generation and reset to 1 if $k = H$.

The third main parameter of DE is the population size ($N$). In the L-SHADE algorithm [17], a relatively simple control strategy has been proposed for $N$, whereby it is initially set to a large value ($N_{max}$) and decreased linearly down to four individuals:

$$N_{g+1} = round\left(\frac{N_{min} - N_{max}}{NFE_{max}} NFE\right) + N_{max}, \tag{8}$$

where $N_{min} = 4$, $NFE$ and $NFE_{max}$ are the current and total available numbers of target function evaluations, and $g$ is the generation number.

The importance of the L-SHADE algorithm is proven by its popularity and the fact that its modifications have been the prize-winning algorithms in many CEC competitions in recent years. Although some other parameter adaptation approaches, such as jDE-based [3], including j100 [18] and j2020 [19], have shown competitive results in some benchmarks, most studies still rely on SHA. Some important modifications of L-SHADE include L-SHADE-RSP [20], which used rank-based selective pressure; DB-LSHADE with distance-based parameter adaptation [21]; and jSO [22], in which heuristic rules were used depending on the current computational resource. One of the recently proposed methods, L-NTADE [7], also used SHA but proposed two populations and modified update strategies. As this method is used as a baseline in this study, it will be considered in more detail.

### 2.3. L-NTADE Algorithm

The L-NTADE [7] algorithm is based on the ideas presented in the Unbounded DE (UDE) in [23], where there is no population size and all previously generated individuals participate in the search process. In L-NTADE. there are two populations: one containing the newest solutions ($x_i^{new}$, $i = 1, \ldots, N$) and the other containing the top solutions ($x_i^{top}$). Both populations have the same size ($N$), and LPSR is applied to them.

The mutation strategy is a modified *current-to-pbest*, with individuals taken from both populations, and is called *r-new-to-ptop/n/t*:

$$v_{i,j} = x_{r1,j}^{new} + F \times (x_{pbest,j}^{top} - x_{i,j}^{new}) + F \times (x_{r2,j}^{new} - x_{r3,j}^{top}), \tag{9}$$

Note that the base vector used to generate a new solution is taken from the newest population ($x_{r1,j}^{new}$) with random index $r1$. Index $r2$ is generated using rank-based selective pressure, whereby ranks are assigned as $rank_i = e^{frac-kp \cdot iN}$, and $kp$ controls the pressure level. Index *pbest* is chosen from one of the $p\%$ best solutions from the top population, and $r3$ is generated randomly with uniform distribution. After mutation, binomial crossover is applied, as well as bound-constraint handling using the midpoint target method.

The selection step in L-NTADE is changed, and imitates the behavior of the unbounded population in UDE; it works as follows:

$$x_{nc} = \begin{cases} u_i, & \text{if } f(u_i) \leq f(x_{r1}^{new}) \\ x_{nc}, & \text{if } f(u_i) > f(x_{r1}^{new}) \end{cases}. \tag{10}$$

where $nc$ is iterated from 1 to $N$ after every successful replacement. In other words, if the trial vector ($u_i$) is better than the base vector ($x_{r1}^{new}$), then an individual with index $nc$ is replaced. Such a strategy may replace a better solution with index $nc$ with a worse solution, leading to continuous updating of the newest population.

In addition to replacing an individual in the newest population, the successful solutions are stored to $x^{temp}$ alongside their fitness values. At the end of the generation, the $x^{top}$ and $x^{temp}$ populations are joined and sorted by fitness, and the best $N$ individuals are saved to $x^{top}$. With this update mechanism, the top population always contains the best $N$ individuals from the whole search process.

### 2.4. Surrogate Modeling and Efficient Global Optimization

The process of solving many real-world problems requires building a mathematical model of the system, which is considered. Gaussian processes, also known as Kriging processes, are a class of models that heavily rely on statistics and Bayesian methods [24]. One of the applications of such models is to optimization problems, in particular, numerical optimization. One of the well-known modern Bayesian optimization methods (originally described in [25]) is the efficient global optimization (EGO), as proposed in [26].

In EGO, the Kriging model is built with a mean function ($\mu$) and a variance function ($\sigma^2$) based on sampling points ($x_1, x_2, \ldots, x_n$) and corresponding outputs ($y_1, y_2, \ldots y_n$). This model acts as a surrogate of the original function to determine the next point where

the target function will be evaluated. There are three main criteria used to determine the best possible position of the next point:

1.  Surrogate-based optimization (SBO), using mean values ($\mu$);
2.  Lower confidence bound (LCB), using the $3\sigma$ confidence interval, i.e., $\mu - 3\sigma$;
3.  Expected improvement (EI), using information about the best known point, as well as $\mu$ and $\sigma$.

EI is usually used in EGO and is calculated as follows:

$$E[I(x)] = E[max(f_{min} - Y, 0)] \tag{11}$$

where $Y$ is a random variable with normal distribution and parameters ($\mu$ and $\sigma$), $f_{min}$ is the best known function value and $E$ determines the expectation. The next point is then calculated as follows:

$$x_{n+1} = \underset{x}{argmax}(E[I(x)]) \tag{12}$$

This step requires solving another optimization problem derived from the original one, and the landscape of $E[I(x)]$ changes after every evaluation. Building a Kriging model and determining $\mu$ and $\sigma$ parameters requires a significant amount of computations, especially if the number of points ($n$) is large or the search space is multidimensional. Therefore, applying EGO makes sense only in expensive cases, i.e., when evaluating the target function ($f(x)$) requires much more time than building a surrogate model and finding the optimum of the expected improvement function.

## 3. Related Work

Despite the significant achievements in developing novel parameter adaptation techniques for DE, there are ways to introduce new approaches by utilizing automated search methods. In general, the parameter adaptation techniques [27] can be divided into two categories [28], namely offline [29,30] and offline [31] adaptation. The difference between these two is that in the offline case, there is a training stage during which knowledge about more efficient schemes is extracted, which is then implemented in a form of a parameter adaptation scheme, which can be used for various types problems or maybe even different algorithms. In the online case, the parameters are tuned during the search, but unlike typical adaptation schemes, which follow predefined strategies, online parameter adaptation tends to extract and utilize knowledge about the problem and the algorithm performance during a single run.

As an example of online parameter adaptation for DE, the method described in [31] can be considered. In this study, the gradients in the *F*- and *Cr*-parameter search space are estimated by copying the population several times and running with different parameters in order to determine the most efficient trajectories. In [6], genetic programming (GP) was applied to design several parameter adaptation techniques for *F* and *Cr* separately, as well as combined strategies. The GP-designed equations, which depended on the current resource, success rate and values from success history adaptation, and different search ranges were considered for the scaling factor (*F*), including negative values. In [32], a similar approach was utilized; instead of genetic programming, the neuroevolution of augmented topologies (NEAT) algorithm was used. Such approaches can be classified as automated design of algorithms (ADA) or genetic improvement (GI) [33].

One main limitation of such hyper-heuristic methods is that they may have limited generalization abilities. That is, if the offline approach is trained on a specific class of problems or specific dimensions or under other limited conditions, then when applied to a new class of problems, such parameter adaptation methods may show limited performance. However, as shown in [6], the heuristics designed using one benchmark may be applicable to others and still show competitive results. Another limitation of using GP or NEAT as offline learners is that these methods require significant computational effort; hence, other techniques can and should be considered.

## 4. Proposed Approach

In the hyper-heuristic approach [34], an algorithm, such as the genetic programming (GP) algorithm, is applied to perform a search for heuristics [35]. The use of GP or a similar approach allows for the automatic design of algorithms (ADA) to be performed or, in other cases, genetic improvement (GI) [33] of existing software in a computation-based framework. The development of this class of algorithms leads to possibilities of automated knowledge extraction and discovery similar to data mining (DM) but for algorithms. In this study, the so-called offline scenario is considered, in which there is a learning phase (searching for hyperparameters), followed by testing and application phases.

In a recent study [6] , the GP for symbolic regression was applied to design new parameter adaptation schemes for $F$ and $Cr$ in differential evolution. One of the findings was that the success rate could be an informative feature to be used for scaling factor adaptation, as some of the symbolic solutions heavily relied on this value. The success rate ($SR$) can be calculated as follows:

$$SR = \frac{NS}{N},\tag{13}$$

where $NS$ is the number of successful solutions, i.e., solutions that were saved during selection (equal to $|S|$). The solution found by GP used the square root of $SR$ to determine scaling factor sampling.

Although the equation found by genetic programming was quite efficient, there was still a possibility of further improvement in terms of efficiency when using $SR$. In order to search for other ways of utilizing the information contained in the success rate, the main idea of the current study was developed. In particular, it was proposed to apply the Taylor series, a well-known universal function approximation tool from calculus. In particular, the 10th-order approximation was applied. The 10th-order was chosen to guarantee the flexibility of the produced curves. In the case of approximating $F$, the following equation is used:

$$MF_r = \sum_{i=1}^{10} c_i (SR - c_0)^i\tag{14}$$

where $MF_r$ is the raw value that determines scaling factor sampling, and $c_i$, $i = 0, 1, \ldots 10$ are the coefficients to be determined. However, this equation may produce very large values, while the scaling factor ($F$) should be within the range of $[0, 1]$. For this purpose, the raw value should be normalized:

$$MF = \frac{(MF_r - MF_{r,min})}{(MF_{r,max} - MF_{r,min})},\tag{15}$$

where $MF_{r,min}$ and $MF_{r,max}$ are the minimum and maximum values, respectively, with $SR \in [0, 1]$, and the $MF$ value is further used as follows:

$$F = randc(MF, 0.1),\tag{16}$$

where $randc(m, s)$ is the Cauchy-distributed random value with location parameter $m$ and scale parameter $s$. As in the SHADE algorithm, if the sampled $F$ is negative or zero, it is sampled again, and if $F > 1$, then it is set to $F = 1$.

The same group of equations was applied to determine the crossover rate, but for sampling, the normal distribution was applied, and $Cr$ values were clipped to the $[0, 1]$ interval.

$$MCr_r = \sum_{i=1}^{10} c_i (SR - c_0)^i\tag{17}$$

$$MCr = \frac{(MCr_r - MCr_{r,min})}{(MCr_{r,max} - MCr_{r,min})},\tag{18}$$

$$Cr = randn(MCr, 0.1).\tag{19}$$

As for the third parameter, population size ($N$), it was determined based on the resource ratio ($RR$):

$$RR = \frac{NFE}{NFE_{max}} \tag{20}$$

The same normalization was applied:

$$MN_r = \sum_{i=1}^{10} c_i (RR - c_0)^i, \tag{21}$$

$$MN = \frac{(MN_r - MN_{r,min})}{(MN_{r,max} - MN_{r,min})}. \tag{22}$$

To determine the population size, the following equation was adapted from LPSR:

$$N_{g+1} = round((N_{min} - N_{max})MN) + N_{max}, \tag{23}$$

Note that increasing the population size is not allowed in this case.

In order to determine the efficiency of the new parameter adaptation scheme, the Mann–Whitney statistical test was applied. In particular, the L-NTADE algorithm, equipped with the Taylor-series-defined curve for $F$, $Cr$ or $N$, was compared to the baseline L-NTADE with standard success–history adaptation. The training phase was performed on the 30 test functions of the CEC 2017 benchmark in the $30D$ case, with 51 independent runs on each function. That is, for every evaluation of the efficiency of $c_i$ coefficients, there were 30 statistical tests performed, one for every test function. Mann–Whitney statistical tests were performed with a significance level of $p = 0.01$, normal approximation and tie breaking. Applying normal approximation is possible because the number of independent runs is large enough (51 runs). Approximating the Mann–Whitney statistics with normal distribution also allowed for calculation of standard score ($Z$) values for every statistical test. If the standard score ($Z$) value was below $-2.58$, the coefficients ($c_i$) for the Taylor series performed significantly worse on a given function than the success–history adaptation. However, if the standard score was larger than 2.58, it performed significantly better. To obtain a smooth target function for coefficient optimization, the total score ($Z_T$) was calculated as a sum over all functions:

$$Z_T = \sum_{j=1}^{30} Z_j, \tag{24}$$

where $Z_j$ is the standard score on the $j$-th function. $Z_T$ values are the target function values for the EGO algorithm applied to search for the Taylor series coefficients ($c_i$), $i = 0, 1, \ldots 10$. One of the disadvantages of such an approach is that $Z_T$ is a random (noisy) value, while the EGO algorithm expects a noiseless target function. A detailed description of the experiments is provided in the next section.

## 5. Experimental Setup and Results

### 5.1. Benchmark Functions and Parameters

To evaluate the possibilities of the proposed approach in searching for parameter adaptation schemes, several experiments were performed, which involved training phases and testing phases. For the training phase, the CEC 2017 Single Objective Bound Constrained Numerical Optimization benchmark functions [8] in the $30D$ case were used. The reason for using $30D$ functions instead of $10D$ is that low-dimensional functions are easier to optimize, and the generalization abilities of the training phase are lower in this case. The computational resources required to learn Taylor series coefficients on $30D$ functions was set to $3 \times 10^5$ function evaluations, as required by the benchmark.

Six learning experiments were performed, in particular for $F$, $Cr$ and $N$, and with two results-ranking methods. In the first ranking method, only the final achieved function value

was considered. In the second case, the ranking described in the CEC 2022 benchmark [9] was used, which also considered convergence speed. In particular, if two or more runs on a test functions were successful, i.e., the goal function value was found, then these runs were compared according to the resources spent on finding the global optimum. Other runs in which the global optimum was not found were ranked as usual. These two ranking types were used in the Mann–Whitney statistical tests. A more detailed description of such a procedure is provided in [36].

When using the CEC 2017 benchmark during the testing phase, dimensions of 10, 30, 50 and 100 were considered. The resource was set to 10,000$D$ evaluations, and 51 independent runs were performed for each of the 30 test functions. The CEC 2022 benchmark contained 12 test functions, with dimensions of 10 and 20, the number of function evaluations set to $2 \times 10^5$ and $1 \times 10^6$ and 30 independent runs.

The following parameters were set for the L-NTADE algorithm: initial population size, $N_{max} = 20D$; minimum population size, $N_{min} = 4$; mutation strategy parameter, $pb = 0.3$; number of memory cells when using SHA, $H = 5$; initial values for memory cells, $M_{F,r} = 0.3$, $M_{Cr,r} = 1.0$, $r = 1, 2, \ldots H$; scaling factor adaptation bias, $pm = 4$; selective pressure parameter for $r2$ index, $kp = 3$. These settings represent a good choice, as demonstrated in [7]. For the EGO algorithm, the search range for $c_i$ values was set to $[-10, 10]$, $i = 0, 1, \ldots 10$, and 1000 target function evaluations were allowed, with an initial set of 25 points generated by a Latin hypercube. The EI criterion was used to determine the next point, and the SLSQP algorithm was applied to search for the optimum in the EI landscape.

The L-NTADE algorithm was implemented in C++ and run on an OpenMPI-powered cluster of 8 AMD Ryzen 3700 PRO devices, with each core each using Linux 20.04. The EGO implementation from the Surrogate Modeling Toolbox (SMT) [37,38] was used in Python 3.8 to determine the Taylor series parameters and automatically run C++ code, as well as Mann–Whitney tests. Results post-processing was also performed using Python 3.8.

*5.2. Numerical Results*

In the first set of experiments, the classical ranking scheme was considered, i.e., the number of function evaluations did not affect the ranking for calculation of $Z_T$ values. Three learning-phase experiments were performed with this setting, namely for $F$, $Cr$ and $N$ parameters. Figure 1 shows the curves found by the EGO by optimizing Taylor series coefficients.

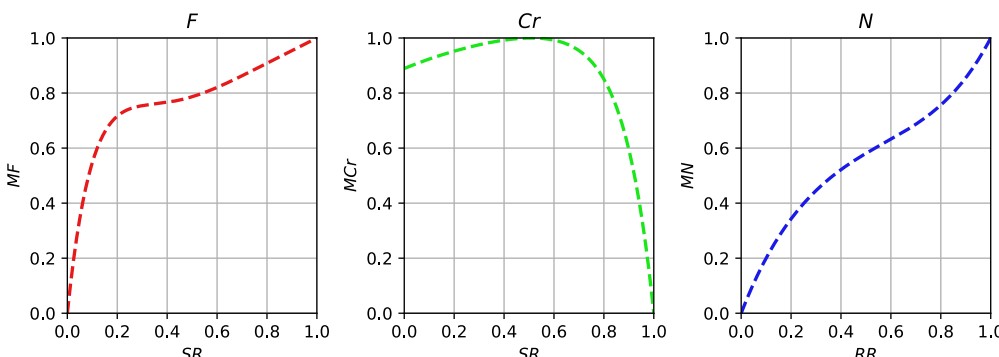

**Figure 1.** Curves for parameter adaptation designed by EGO for Taylor series and the best function values used in ranking.

The curve found for the scaling factor parameter ($F$) shown in Figure 1 has a specific shape, with a fast growth from 0 to around 0.7 when the success rate $SR$ changes from 0 to 0.2, after which the curve is less steep. Note that for the most of the time during the search, the success rate stays rather low (lower than 0.5), which means that $MF$ changes between 0 and 0.8. The curve found for the crossover rate ($Cr$) is different, starting from 0.9 and increasing to 1.0 around $SR = 0.5$. This means that most of the time, large $Cr$ values

are used, with little dependence on the success rate. For the population size, the curve for *MN* values is different from a classical linear curve and has an inflection point around 0.5. In particular, during the early phase of the search, when *RR* is small and the population is large, its decrease becomes more rapid than linear, while during later phases of the search, the population remains larger for a longer period of time, eventually decreasing more rapidly to promote exploitation at the very end of the search.

Table 1 contains a performance comparison of the designed heuristics applied to L-NTADE separately and altogether. Here, the best function value was used for ranking in Mann–Whitney tests.

**Table 1.** Mann–Whitney tests of L-NTADE against versions with designed heuristics, CEC 2017 benchmark, number of wins/ties/losses and total standard score.

| Algorithm | $10D$ | $30D$ | $50D$ | $100D$ |
|---|---|---|---|---|
| L-NTADE$_{MF}$ vs. L-NTADE | 9/16/5 (32.37) | 15/13/2 (72.67) | 11/14/5 (36.05) | 12/10/8 (25.54) |
| L-NTADE$_{MCr}$ vs. L-NTADE | 0/29/1 (−5.39) | 0/30/0 (6.44) | 2/25/3 (−15.71) | 0/26/4 (−28.30) |
| L-NTADE$_{MN}$ vs. L-NTADE | 0/30/0 (−14.23) | 0/30/0 (8.14) | 2/28/0 (3.53) | 2/26/2 (1.73) |
| L-NTADE$_{MF,MCr,MN}$ vs. L-NTADE | 7/15/8 (12.38) | 10/18/2 (42.70) | 7/15/8 (−5.91) | 8/7/15 (−15.53) |

As shown in Table 1, applying scaling factor adaptation based on success rate with coefficients found by EGO produces much better results in all dimensions. It is worth noting that the largest effect is on $30D$ functions, as this is where the training was performed; however, in other dimensions, there are also significant improvements on many functions. Applying designed heuristics for *Cr* does not produce good results, except for a small improvement in the $30D$ case. In other dimensions, the efficiency is decreased. The population size control strategy has similar performance to the standard LPSR, with slight differences in the $50D$ and 100 cases. When all three are combined, the overall efficiency drops compared to using only one strategy for scaling factor adaptation.

Table 2 contains a comparison on the CEC 2022 benchmark set. Here, the ranking in the Mann–Whitney test also considers the computational resources spent to find the optimum.

**Table 2.** Mann–Whitney tests of L-NTADE against versions with designed heuristics, CEC 2022 benchmark, number of wins/ties/losses and total standard score.

| Algorithm | $10D$ | $20D$ |
|---|---|---|
| L-NTADE vs. L-NTADE$_{MF}$ | 5/4/3 (24.29) | 3/5/4 (−12.66) |
| L-NTADE vs. L-NTADE$_{MCr}$ | 3/9/0 (16.02) | 3/9/0 (14.97) |
| L-NTADE vs. L-NTADE$_{MN}$ | 6/6/0 (32.05) | 3/9/0 (13.99) |
| L-NTADE vs. L-NTADE$_{MF,MCr,MN}$ | 5/4/3 (22.87) | 2/6/4 (-11.28) |

The results in Table 2 show that the heuristic for scaling factor adaptation performs better than SHA in the $10D$ case but worse in the $20D$ case. However, applying designed curves for crossover rate and population size results increased efficiency in both dimensions. When all three are combined, the results are similar to those achieved with *MF* only. It should be noted that most of the improvements observed when using *MCr* and *MN* were in terms of the number of function evaluations required to find a solution. In other words, the algorithm converged faster. This demonstrates that setting a large *Cr* value for L-NTADE when solving CEC 2022 benchmark functions could be more efficient than using success–history adaptation and that the linear population size reduction may be not the best option. The last statement is also supported by the fact that the NL-SHADE-RSP [39]

and NL-SHADE-LBC [40] algorithms, which were one of the top methods in the CEC 2021 and CEC 2022 benchmarks, also used non-linear population size reduction.

Figure 2 demonstrates the curves designed by efficient global optimization when the computational resources were taken into consideration during ranking.

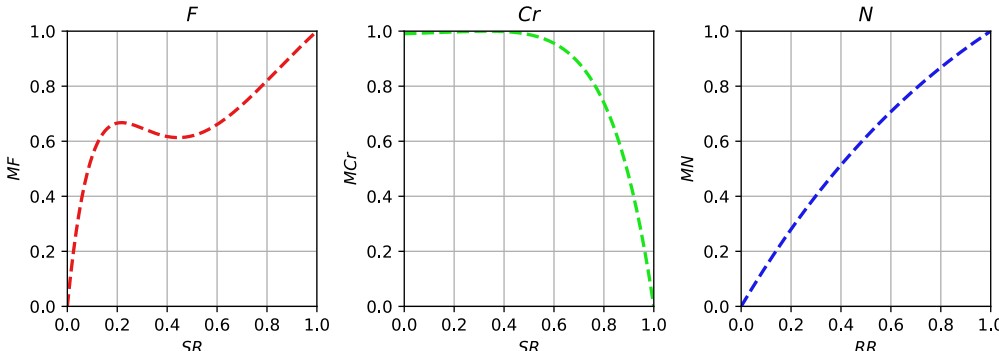

**Figure 2.** Curves for parameter adaptation designed by EGO for Taylor series, the best found function values and spent resources used in ranking.

The curves shown in Figure 2 are quite similar to those shown in Figure 1, although some differences can be observed. First of all, the curve for $F$ is lower, although the steepness is similarly close to zero. At the same time, the curvature around 0.4 is more expressed, with $MF$ decreasing, with a minimum point around 0.5. Applying such a curve would result in smaller $F$ values being generated compared to the previous case if the success rate is higher. The curve for the crossover rate ($Cr$) is quite simple and sets $MCr$ to 1, independent of the $SR$ value, which rarely reached values above 0.5 during the active phase of the search. For the population size control parameter ($MN$), the designed curve is similar to that of non-linear population size reduction, i.e., it reduces the population size more rapidly so that more computations are performed with a smaller population size.

Table 3 contains a comparison of the designed heuristics for parameter adaptation on the CEC 2017 benchmark.

**Table 3.** Mann–Whitney tests of L-NTADE against versions with designed heuristics, CEC 2017 benchmark, number of wins/ties/losses, total standard score and learning considering the convergence speed.

| Algorithm | 10$D$ | 30$D$ | 50$D$ | 100$D$ |
|---|---|---|---|---|
| L-NTADE$_{MF}$ vs. L-NTADE | 8/16/6 (25.43) | 14/15/1 (71.80) | 13/14/3 (52.68) | 12/15/3 (67.00) |
| L-NTADE$_{MCr}$ vs. L-NTADE | 0/28/2 (−7.57) | 2/28/0 (−5.41) | 2/25/3 (−15.76) | 2/19/9 (−26.57) |
| L-NTADE$_{MN}$ vs. L-NTADE | 0/29/1 (−13.89) | 1/28/1 (5.95) | 0/29/1 (−11.06) | 1/27/2 (−12.43) |
| L-NTADE$_{MF,MCr,MN}$ vs. L-NTADE | 10/15/5 (19.05) | 9/18/3 (49.66) | 8/16/6 (16.65) | 11/9/10 (11.79) |

The first row in Table 3 shows that the heuristic designed by EGO performs much better than the standard success–history adaptation. Moreover, compared to the results presented in Table 1, it achieves better generalization, as it performs much better in the 50$D$ and 100$D$ cases. The heuristic for the crossover rate adaptation does not produce any good results, and is always worse than SHA. This means that the success rate ($SR$) is not important for $Cr$. The control method for population size reduction works similarly in the 30$D$ case but loses performance in all other dimensions. Finally, combining all three methods still increases performance but not as significantly because the scaling factor

parameter (*F*) adaptation is held back by two other inefficient methods. Note that here the standard ranking scheme was performed, although the spent computational resources were taken into consideration during training. This may explain the worse results obtained when searching for *MCr* and *MN* curves.

Table 4 contains a comparison on the CEC 2022 benchmark.

**Table 4.** Mann–Whitney tests of L-NTADE against versions with designed heuristics, CEC 2022 benchmark, number of wins/ties/losses, total standard score and learning considering convergence speed.

| Algorithm | 10*D* | 20*D* |
|:---:|:---:|:---:|
| L-NTADE vs. L-NTADE$_{MF}$ | 6/3/3 (25.21) | 4/5/3 (10.62) |
| L-NTADE vs. L-NTADE$_{MCr}$ | 5/7/0 (27.65) | 3/8/1 (8.20) |
| L-NTADE vs. L-NTADE$_{MN}$ | 6/6/0 (31.07) | 3/9/0 (9.21) |
| L-NTADE vs. L-NTADE$_{MF,MCr,MN}$ | 6/3/3 (27.43) | 4/4/4 (9.00) |

In case of the CEC 2022 benchmark, where the amount of computational resources spent on finding the optimum was taken into consideration, the designed heuristics for *Cr* and *N* performed much better. The curve designed for the scaling factor still performed better in both dimensions. However, combining all three together did not result in any significant benefits.

Figures 3 and 4 demonstrate the process of parameter adaptation as observed on some of the 30*D* functions of the CEC 2017 benchmark. The three shown curves are the success rate (*SR*) value for *F* sampling *MF*, corresponding to the value of all memory cells when the SHA was applied. In all cases, the heuristics for *Cr* and *N* were disabled.

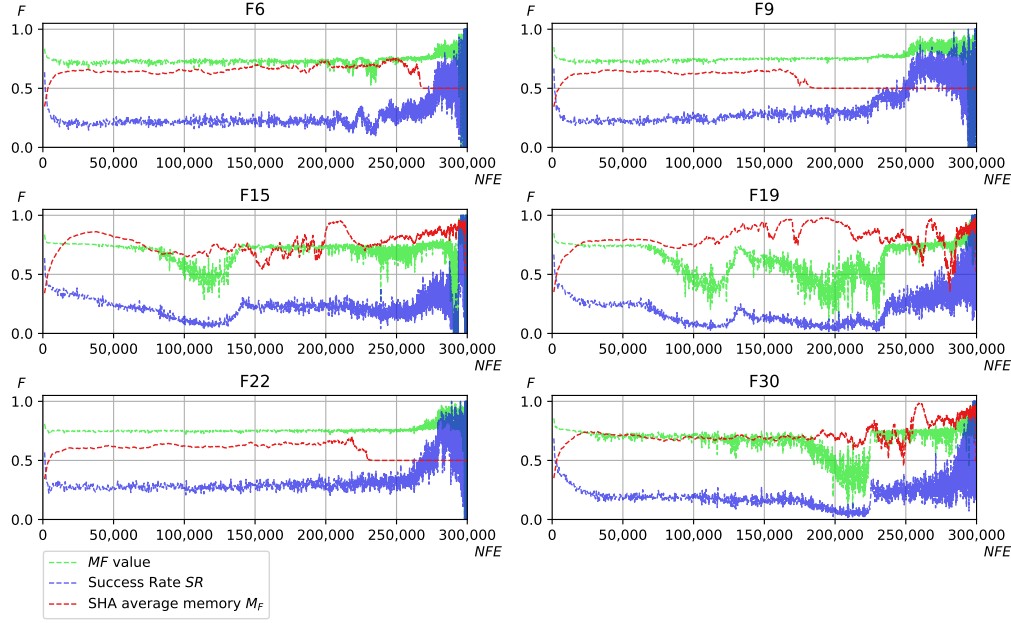

**Figure 3.** Scaling factor adaptation process, coefficients found with standard ranking, SHA adaptation for comparison, CEC 2017 and selected functions.

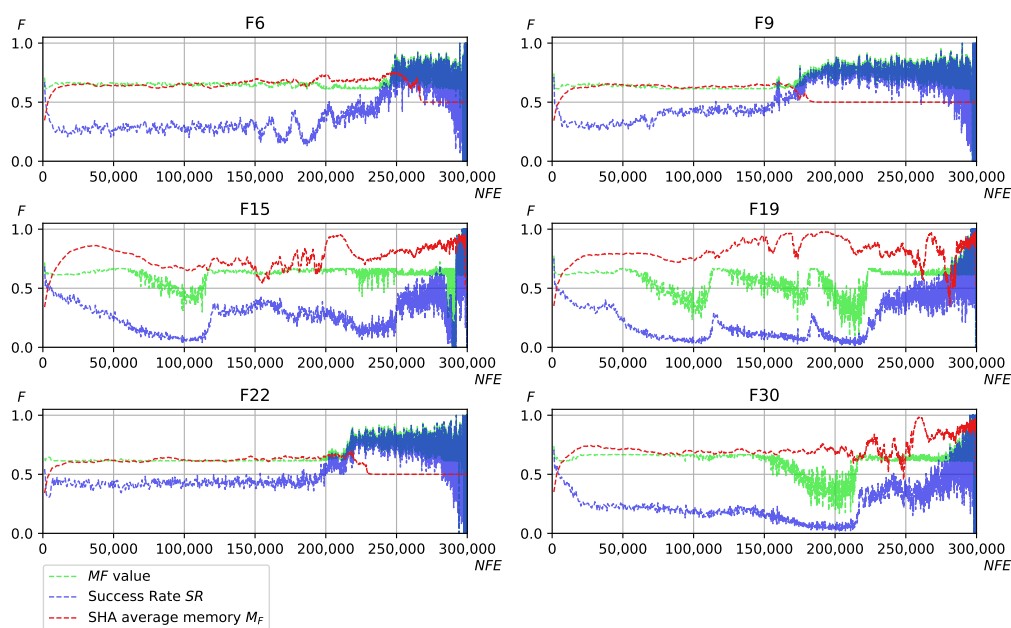

**Figure 4.** Scaling factor adaptation process, coefficients found with ranking by spent resources, SHA adaptation for comparison, CEC 2017 and selected functions.

As shown in Figures 3 and 4, for the first heuristic designed by EGO when the standard ranking was used in Mann–Whitney tests, the *MF* values are quite high and, in many cases, higher than those generated by SHA. In the second case, when the computational resources were considered during the training phase, the designed heuristic resulted in very similar *F* values as SHA, for example, on functions 6, 9, 22 and 30. In other cases, however, there were significant differences. Observing the dependence between success rate and *MF*, one may notice that when *SR* drops low enough, the *MF* values start oscillating, as they appear to be on the steep part of the curve. This results in a more broad search, i.e., more diverse *F* values are generated. If the success rate is relatively high, i.e., the search is successful, the *MF* values are more stable.

The coefficients found for all the curves in Figures 1 and 2 are provided in Table 5.

**Table 5.** Coefficients found by EGO for Taylor series.

| Ranking | Best Value | | | Best Value and Spent Resources | | |
|---|---|---|---|---|---|---|
| **Parameter** | ***F*** | ***Cr*** | ***N*** | ***F*** | ***Cr*** | ***N*** |
| $c_0$ | 1.063 | −0.073 | 0.453 | 1.057 | −0.766 | 10.000 |
| $c_1$ | 9.727 | 10.000 | 3.791 | 10.000 | 10.000 | −10.000 |
| $c_2$ | 3.228 | −8.230 | −3.924 | −10.000 | −10.000 | 10.000 |
| $c_3$ | 10.000 | 2.472 | 10.000 | −10.000 | 10.000 | 10.000 |
| $c_4$ | 5.805 | 5.290 | 1.721 | 10.000 | 5.988 | −10.000 |
| $c_5$ | −10.000 | −10.000 | 10.000 | −10.000 | 4.252 | −10.000 |
| $c_6$ | 8.923 | −10.000 | −10.000 | −3.236 | 3.455 | −4.892 |
| $c_7$ | 3.450 | 10.000 | −10.000 | 5.773 | 4.214 | 10.000 |
| $c_8$ | −3.375 | 10.000 | 10.000 | −2.913 | −4.950 | 9.921 |
| $c_9$ | 5.084 | −10.000 | 10.000 | −2.647 | −0.346 | 9.504 |
| $c_{10}$ | −8.966 | −8.535 | 5.942 | −9.479 | −3.835 | −9.717 |

To compare the proposed approach with alternative methods, in Tables 6 and 7, the L-NTADE with scaling factor adaptation method found in the second scenario (with spent computational resources considered, Figure 2) is used. This version was chosen as it performed better than any other version. For comparison on CEC 2017 benchmark, the results of some of the top methods from CEC 2017 and CEC 2018 competitions were chosen.

**Table 6.** Mann–Whitney tests of L-NTADE$_{MF}$ against other approaches, CEC 2017, number of wins/ties/losses and total standard score.

| Algorithm | 10$D$ | 30$D$ | 50$D$ | 100$D$ |
|---|---|---|---|---|
| L-NTADE$_{MF}$ vs. LSHADE-SPACMA [41] | 7/18/5 (9.88) | 17/6/7 (67.92) | 14/3/13 (18.56) | 14/1/15 (−4.52) |
| L-NTADE$_{MF}$ vs. jSO [22] | 5/20/5 (−7.43) | 19/10/1 (131.06) | 22/5/3 (149.74) | 24/1/5 (145.89) |
| L-NTADE$_{MF}$ vs. EBOwithCMAR [42] | 2/17/11 (−49.47) | 16/9/5 (73.42) | 18/7/5 (101.44) | 20/4/6 (115.96) |
| L-NTADE$_{MF}$ vs. L-SHADE-RSP [20] | 4/20/6 (−12.29) | 18/11/1 (116.89) | 20/7/3 (126.03) | 20/4/6 (120.30) |
| L-NTADE$_{MF}$ vs. NL-SHADE-RSP [39] | 13/6/11 (9.29) | 24/3/3 (173.75) | 28/2/0 (246.33) | 29/0/1 (233.83) |
| L-NTADE$_{MF}$ vs. NL-SHADE-LBC [40] | 4/20/6 (−12.52) | 24/5/1 (180.53) | 28/2/0 (228.70) | 27/2/1 (215.99) |
| L-NTADE$_{MF}$ vs. L-NTADE [7] | 8/16/6 (25.43) | 14/15/1 (71.80) | 13/14/3 (52.68) | 12/15/3 (67.00) |

Comparing L-NTADE$_{MF}$ to other methods, one may conclude that it performs much better, especially in high-dimensional cases, compared to L-SHADE-RSP, jSO and EBOwith-CMAR, although it loses to LSHADE-SPACMA in the 100$D$ case. In the 10$D$ case, it works worse than EBOwithCMAR and L-SHADE-RSP, but the performance improvement in other dimensions is much more significant.

For the CEC 2022 benchmark, the top-three best methods were chosen for comparison, as well as some other algorithms.

**Table 7.** Mann–Whitney tests of L-NTADE$_{MF}$ against the top-three competition and other approaches, CEC 2022, number of wins/ties/losses and total standard score.

| Algorithm | 10$D$ | 20$D$ |
|---|---|---|
| APGSK-IMODE [43] | 8/2/2 (35.92) | 8/2/2 (43.60) |
| MLS-LSHADE [44] | 7/2/3 (28.25) | 5/2/5 (−1.21) |
| MadDE [45] | 8/2/2 (40.34) | 7/2/3 (28.96) |
| EA4eigN100 [46] | 5/0/7 (−5.93) | 5/2/5 (0.42) |
| NL-SHADE-RSP-MID [47] | 5/3/4 (8.55) | 6/2/4 (13.75) |
| L-SHADE-RSP [20] | 6/3/3 (20.02) | 5/4/3 (9.04) |
| NL-SHADE-RSP [39] | 7/2/3 (28.85) | 7/2/3 (26.33) |
| NL-SHADE-LBC [40] | 7/3/2 (30.05) | 4/5/3 (8.48) |
| L-NTADE [7] | 6/3/3 (25.21) | 4/5/3 (10.62) |

As shown in Table 7, the L-NTADE$_{MF}$ has higher efficiency than most methods, except EA4eigN100, which is better in the 10$D$ case and achieves almost the same performance in the 20$D$ case. Note that L-NTADE$_{MF}$ has the same parameter settings for both benchmarks, so it was not tuned for CEC 2022, which requires considerably more computational resources. Applying the designed heuristic for scaling factor sampling still produces better results compared to most of the algorithms.

In order to evaluate the generalization abilities of the designed parameter adaptation techniques, several additional experiments were performed. In particular, as in [6], a similar offline approach using genetic programming was considered. The results of this study were used for comparison. The NL-SHADE-RSP algorithm equipped with five different automatically designed heuristics was compared with the same NL-SHADE-RSP algorithm with success-rate-based adaptation of scaling factor, where Taylor series coefficients are taken from the second experiment, i.e., with spent resources considered (Figure 2). Table 8

shows a comparison of NL-SHARE-RSP$_{MF}$ with the results from [6] on the CEC 2017 benchmark, where different GP-designed heuristics are marked with parameters that are tuned (*F*, *Cr* or both), as well as the allowed range ([0, 1] or [−1.2, 1.2]).

**Table 8.** Mann–Whitney tests of the best designed heuristics from [6] against NL-SHADE-RSP$_{MF}$, CEC 2017 and number of wins/ties/losses.

| Dimension | $F[0, 1]$ | $Cr[0, 1]$ | $F[-1.2, 1.2]$ |
|:---:|:---:|:---:|:---:|
| 10 | 7/20/3 | 3/23/4 | 6/21/3 |
| 30 | 21/7/2 | 22/6/2 | 21/7/2 |
| 50 | 26/4/0 | 27/3/0 | 25/5/0 |
| 100 | 28/1/1 | 25/4/1 | 26/3/1 |
| Total | 82/32/6 | 77/36/7 | 78/36/6 |
| **Dimension** | $F[0, 1], Cr[0, 1]$ | $F[-1.2, 1.2], Cr[0, 1]$ | |
| 10 | 4/22/4 | 4/22/4 | |
| 30 | 20/8/2 | 20/8/2 | |
| 50 | 27/2/1 | 25/5/0 | |
| 100 | 26/3/1 | 27/2/1 | |
| Total | 77/35/8 | 76/37/7 | |

As shown in Table 8, the NL-SHADE-RSP algorithm with the heuristic designed in this study using Taylor series and efficient global optimization wins against all the heuristics proposed by genetic programming, especially in high-dimensional cases. Also note that the curve parameters of the Taylor series expansion were designed for the L-NTADE algorithm, which has a significantly different structure. However, the same parameters enabled the same highly competitive results in NL-SHADE-RSP. This means that the designed parameter adaptation technique has generalization capabilities.

To further test the proposed method, the NL-SHADE-RSP$_{MF}$ algorithm was tested on the SOCO benchmark set. Unlike the CEC benchmarks, the SOCO test suite [10] has problems with 50, 100, 200, 500 and 1000 dimension variables. The 1000*D* case was not considered because it was not considered in [6], with no results for some of the methods for this setting. Fewer computational resources were spent on SOCO (only 5000*D* evaluations), and only 19 test functions were considered. Table 9 compares the NL-SHADE-RSP$_{MF}$ against NL-SHADE-RSP with the best GP-designed heuristic, i.e., $F[-1.2, 1.2], Cr[0, 1]$ as well as other approaches.

**Table 9.** Comparison of NL-SHADE-RSP$_{MF}$ with other approaches on the SOCO benchmark and the number of wins/ties/losses.

| Algorithm | 50*D* | 100*D* |
|:---:|:---:|:---:|
| DE [11] | 9/6/4 | 9/6/4 |
| CHC [48] | 19/0/0 | 19/0/0 |
| G-CMA-ES [49] | 15/2/2 | 14/1/4 |
| SOUPDE [50] | 7/7/5 | 8/6/5 |
| DE-D$^{\wedge}$40 + M$^{\wedge}$m [51] | 10/4/5 | 11/3/5 |
| GODE [52] | 9/6/4 | 8/6/5 |
| GaDE [53] | 5/9/5 | 2/9/8 |
| jDElscop [54] | 4/10/5 | 4/9/6 |
| SaDE-MMTS [55] | 3/10/6 | 1/10/8 |
| MOS [56] | 4/11/4 | 2/10/7 |
| MA-SSW-Chains [57] | 15/1/3 | 15/0/4 |

**Table 9.** *Cont.*

| Algorithm | 50$D$ | 100$D$ |
|---|---|---|
| RPSO-vm [58] | 12/3/4 | 11/4/4 |
| Tuned IPSOLS [59] | 11/4/4 | 7/4/8 |
| EvoPROpt [60] | 17/0/2 | 17/0/2 |
| EM323 [61] | 11/4/4 | 10/4/5 |
| VXQR1 [62] | 10/6/3 | 9/4/6 |
| NL-SHADE-RSP$_{F[-1.2,1.2],Cr[0,1]}$ [6] | 4/12/3 | 6/11/2 |
| Total | 165/95/63 | 153/87/83 |
| **Algorithm** | **200$D$** | **500$D$** |
| DE [11] | 4/6/9 | 3/4/12 |
| CHC [48] | 19/0/0 | 19/0/0 |
| G-CMA-ES [49] | 13/2/4 | 14/1/4 |
| SOUPDE [50] | 6/4/9 | 4/3/12 |
| DE-D$^\wedge$40 + M$^\wedge$m [51] | 8/2/9 | 5/2/12 |
| GODE [52] | 4/6/9 | 3/4/12 |
| GaDE [53] | 3/6/10 | 2/4/13 |
| jDElscop [54] | 2/7/10 | 1/5/13 |
| SaDE-MMTS [55] | 0/8/11 | 1/6/12 |
| MOS [56] | 1/8/10 | 1/6/12 |
| MA-SSW-Chains [57] | 14/0/5 | 10/0/9 |
| RPSO-vm [58] | 9/4/6 | 9/1/9 |
| Tuned IPSOLS [59] | 7/4/8 | 4/3/12 |
| EvoPROpt [60] | 15/0/4 | 11/1/7 |
| EM323 [61] | 8/4/7 | 8/2/9 |
| VXQR1 [62] | 9/3/7 | 9/2/8 |
| NL-SHADE-RSP$_{F[-1.2,1.2],Cr[0,1]}$ [6] | 6/8/5 | 8/5/6 |
| Total | 128/72/123 | 112/49/162 |

Table 9 shows that the NL-SHADE-RSP algorithm with the parameter adaptation technique designed by EGO with computational resource consideration outperforms most of the algorithms in the 50$D$ and 100$D$ cases but loses to some of the algorithms in 200$D$ and 500$D$ cases. Considering that the parameters were designed for the 30$D$ case and a different algorithm L-NTADE on a different benchmark set, this can be seen as a highly competitive result. As for the comparison to the GP-designed heuristic, the proposed algorithm shows better efficiency in all cases, i.e., it performs better on most test functions in all dimensions.

## 6. Discussion

The experimental results presented in the previous section demonstrate that the proposed approach, i.e., using Taylor series and efficient global optimization, can be successfully applied to design new heuristics for evolutionary algorithms. Such a hyper-heuristic approach appears to be quite universal, as it can be applied to almost any algorithm, where some dependence between a couple (or more) parameters should be derived, and it is not clear what it should be. Moreover, it can be used to discover whether there is any dependence that could be utilized at all. For example, we showed that there is no sense in attempting to determine the crossover rate based on the success rate, while the scaling factor depends on it significantly. The disadvantage of such an approach is its computational burden. Each of the six experiments performed within this study required around 24 h to complete 1000 evaluations, and a cluster for parallel evaluations was is use. Nevertheless, this framework can be applied to discover new ways to tune and improve evolutionary algorithms.

The most efficient heuristic found in this study, *MF* from the second setting with computational resource consideration during the learning phase, exhibits similar trends to those observed when genetic programming was applied for parameter adaptation method

design in [6]. The GP showed that a square root for success rate can be used, and using Taylor series revealed a similar dependence: a steep increase from 0 and a more flat increase after 0.4. It is important to understand why this simple strategy works and, in many cases, works better than classical success–history adaptation. We hypothesize that it works efficiently due to its combination with the *current-to-pbest/1* strategy. If the success rate is relatively high (close to 0.4), then larger $F$ values should be set, which would introduce new solutions closer to one of the $p\%$ best solutions, promoting exploitation. If, however, the search is inefficient, then the $F$ value should be decreased, as moving towards one of the $p\%$ best solutions does not lead to better results. In this case, smaller $F$ values should be sampled, with larger oscillations of $MF$ value, resulting in a more diverse and broad search, i.e., exploration. In this manner, applying a simple curve and describing the dependence between the success rate and scaling factor leads to logical behavior of the algorithm, improving overall performance. The SHA, on the other hand, attempts to catch possible improvements, which may sometimes be difficult, as these improvements may be rare, resulting in excessively high memory cell values, as observed in the graphs.

Possible directions for further studies combining DE and EGO with Taylor series approximation include:

1. Experimentally searching for an efficient information source for $Cr$ adaptation;
2. Applying multidimensional Taylor series to design parameter adaptation based on more than one feature;
3. Searching for an efficient control strategy for the $pb\%$ parameter;
4. Applying the described approach to other DE algorithms;
5. Experimenting with search methods other than EGO.

The described approach can be applied to other areas of evolutionary computation.

## 7. Conclusions

In this study, we proposed a surrogate-assisted approach to design novel parameter adaptation heuristics for differential evolution by describing the dependence between different values using Taylor series. The described approach applied to a recently proposed L-NTADE algorithm achieved significant performance improvements in several cases, especially with scaling factor adaptation. Efficient global optimization enabled parameter tuning for the Taylor-series-defined curve with a relatively small number of evaluations. The resulting modified algorithm demonstrated its high efficiency compared to alternative approaches on two benchmarks, proving is workability. The most important part is that the described approach is universal and can be utilized to improve other algorithms in a hyper-heuristic framework.

**Author Contributions:** Conceptualization, V.S. and E.S.; methodology, V.S. and E.S.; software, V.S.; validation, V.S. and E.S.; formal analysis, V.S.; investigation, V.S.; resources, E.S. and V.S.; data curation, E.S.; writing—original draft preparation, V.S.; writing—review and editing, V.S.; visualization, V.S.; supervision, E.S.; project administration, E.S.; funding acquisition, E.S. and V.S. All authors have read and agreed to the published version of the manuscript.

**Funding:** This work was supported by the Ministry of Science and Higher Education of the Russian Federation within limits of state contract № FEFE-2023-0004.

**Data Availability Statement:** Not applicable.

**Conflicts of Interest:** The authors declare no conflict of interest.

## Abbreviations

The following abbreviations are used in this manuscript:

| | |
|---|---|
| GA | Genetic algorithm |
| GP | Genetic programming |
| EC | Evolutionary computation |
| DE | Differential evolution |
| NEAT | Neuroevolution of augmented topologies |
| EGO | Efficient global optimization |
| CEC | Congress on Evolutionary Computation |
| SHADE | Success–history adaptive differential evolution |
| LPSR | Linear population size reduction |
| LBC | Linear bias change |
| RSP | Rank-based Selective pressure |
| UDE | Unbounded differential evolution |
| L-NTADE | Linear population size reduction Newest and Top Adaptive Differential Evolution |

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
