# Peer review of "Surrogate-Assisted Automatic Parameter Adaptation Design for Differential Evolution"

_mathematics, doi:10.3390/math11132937_

Round 1

Reviewer 1 Report

The author of the paper propose a method for automatic adaptation of DE algorithm parameters. They pretend that their approach is efficient and uses small computational resources. The paper is well written and the proposed algorithm and ideas are are described in comprehensive way and can be repeated . The weakness of the paper are tests.

It is not clear which benchmark tests are used, all or part of them. Other weakness is the dimension of the CEC 2022 benchmark problems, 10 and 20. They need to be minimum 100.

Author Response

Review #1:

The author of the paper propose a method for automatic adaptation of DE algorithm parameters. They pretend that their approach is efficient and uses small computational resources. The paper is well written and the proposed algorithm and ideas are are described in comprehensive way and can be repeated . The weakness of the paper are tests.

Answer: Thank you for your valuable comments. We have revised the paper, adding new subsection, experiments and references. All changes are highlighted in blue.

It is not clear which benchmark tests are used, all or part of them. Other weakness is the dimension of the CEC 2022 benchmark problems, 10 and 20. They need to be minimum 100.

Answer: We have performed additional experiments on the SOCO benchmark suite (with up to 500 variables) applying the desinged heuristics to a different algorithm, NL-SHADE-RSP. So, there are now 3 benchmark tests used, CEC 2017 with all four dimensions: 10, 30, 50, 100; CEC 2022 with all two dimensions: 10, 20 and SOCO with four dimensions: 50, 100, 200, 500.

Reviewer 2 Report

The authors propose a methodology (hyper-heuristic) to adapt the three parameters required in the optimization method called Differential Evolution (DE). This is done to improve convergence to the pseudo-optimal solution.

The contribution is not cleared stated in the introduction. The authors should not confuse the expected results of the study with its contributions.

Section 2 is more about the existing methods that are used in the study. It is not about related works. The paper must have both. The current contents of SEction 2 describes the existing  methods on which the work is based. It should be titled some thing like "Background". The authors must include a new section title "Related works" that reviews existing research work that are baout the same problem treated in this paper, which is how to adapt/find automatically the parameter setting in DE and similar optimization methods. The authors shoudl also contrast the proposed methods with these existing related ones to establish the real contributions of the proposed work.

Regarding the performance results, the presented comparison is poor. Comparision of the achieved results must be compared to those obtained when using state-of-the-art approaches, as described in the Related Works section (to be added). 

So, the paper must go through a major revision: the contribution must be made cristal clear in the introduction; a new section about state-of-the-art existing related works on automatic parameter adataption for stochastic optimization methods must be added. THis ental a bibliography research and the reference section would include many new reference on the subject; the comparison must be improved to establish the superiority of the proposed techniques.

The english used is fine.

Author Response

Review #2:

The authors propose a methodology (hyper-heuristic) to adapt the three parameters required in the optimization method called Differential Evolution (DE). This is done to improve convergence to the pseudo-optimal solution.

The contribution is not cleared stated in the introduction. The authors should not confuse the expected results of the study with its contributions.

Answer: Thank you for your valuable comments. We have revised the contributions part, and added additional item. All changed parts are highlighted in blue.

Section 2 is more about the existing methods that are used in the study. It is not about related works. The paper must have both. The current contents of SEction 2 describes the existing  methods on which the work is based. It should be titled some thing like "Background". The authors must include a new section title "Related works" that reviews existing research work that are baout the same problem treated in this paper, which is how to adapt/find automatically the parameter setting in DE and similar optimization methods. The authors shoudl also contrast the proposed methods with these existing related ones to establish the real contributions of the proposed work.

Answer: We have added a new subsection 2.4 describing some of the existing studies on offline and online methods for parameter adaptation in DE.

Regarding the performance results, the presented comparison is poor. Comparision of the achieved results must be compared to those obtained when using state-of-the-art approaches, as described in the Related Works section (to be added). 

Answer: We have performed additional experiments comparing the designed heuristics with those generated by genetic programming in a different study. Also, we used another benchmark and applied designed heuristics to a different algorithm to prove the generalization abilities.

So, the paper must go through a major revision: the contribution must be made cristal clear in the introduction; a new section about state-of-the-art existing related works on automatic parameter adataption for stochastic optimization methods must be added. THis ental a bibliography research and the reference section would include many new reference on the subject; the comparison must be improved to establish the superiority of the proposed techniques.

Answer: Thank you again for your comments. We tried our best by adding more description of similar methods, references and experiments. All changes in the paper are highlighted in blue.

Round 2

Reviewer 1 Report

The author taken into account all reviewers comments and remarks. The paper can be published as it is.

Author Response

The author taken into account all reviewers comments and remarks. The paper can be published as it is.

Answer: Thank you for good evaluation of our work. We have made minor changes requested by other reviewer.

Reviewer 2 Report

I still think that the added subsection 2.4 should be a separate section titled as "Related works" and the rest of Section 2 should be used to form a separate section titled "Background" or something similar.

Figures 3 and 4 have nor color legend. This must be corrected in the final version.

proofreading is necessary.

Author Response

I still think that the added subsection 2.4 should be a separate section titled as "Related works" and the rest of Section 2 should be used to form a separate section titled "Background" or something similar.

Answer: Thank you for your valuable comments. We have moved subsection 2.4 into a separate section "Related work", and the rest of Section 2 is now called "Background".

Figures 3 and 4 have nor color legend. This must be corrected in the final version.

Answer: The legend was present on the graph for F30, we have moved it under all graphs for better visibility.